# A TKA Insert with A Lateral Flat Articular Surface Maximizes External and Internal Tibial Orientations without Anterior Lift-Off Relative to Low- and Ultracongruent Surfaces

**DOI:** 10.3390/jpm12081274

**Published:** 2022-08-03

**Authors:** Alexander J. Nedopil, Stephen M. Howell, Maury L. Hull

**Affiliations:** 1Orthopädische Klinik König-Ludwig-Haus, Lehrstuhl für Orthopädie der Universität Würzburg, 97074 Wurzburg, Germany; 2Department of Biomedical Engineering, University of California, Davis, CA 95616, USA; sebhowell@mac.com (S.M.H.); mlhull@ucdavis.edu (M.L.H.); 3Department of Mechanical Engineering, University of California, Davis, CA 95616, USA; 4Department of Orthopedic Surgery, University of California Davis Medical Center, Sacramento, CA 95817, USA

**Keywords:** total knee arthroplasty, kinematic alignment, implant design, PCL retention, congruency congruency

## Abstract

Background: In total knee arthroplasty (TKA), inserts can have different levels of medial and lateral congruency determined by the acuteness of the upslopes of the anterior and posterior articular surfaces. The present study evaluated an insert with different levels of lateral congruency and a medial ball-in-socket congruency to test the hypothesis that a lateral flat (F) insert maximizes external tibial orientation at extension and internal orientation at 90° flexion and lowers the incidence of anterior lift-off relative to low-congruent (LC) and ultracongruent (UC) lateral inserts. Methods: Two surgeons treated 23 patients with unrestricted caliper-verified kinematic alignment (KA) and posterior cruciate ligament (PCL) retention. They randomly trialed inserts with a medial radial dial that functioned as a built-in goniometer by measuring the tibial orientation relative to a sagittal line on the femoral trial component. Anterior lift-off of the insert from the baseplate indicated PCL tightness. Results: The F insert’s mean of 9° of external tibial orientation was higher than that of the LC (5°, *p* < 0.0001) and UC inserts (2°, *p* < 0.0001). The −13° of internal tibial orientation at 90° flexion was higher than that of the LC (−9°, *p* < 0.0001) and UC inserts (−7°, *p* < 0.0001). The 0% incidence of anterior lift-off was less than that of the LC (26%) and UC inserts (57%) (*p* < 0.0001). Conclusions: Surgeons and implant manufacturers should know that adding congruency to the lateral articular surface limits external tibial orientation in extension and internal tibial orientation at 90° flexion and overtightens the PCL. These rotational limitations and flexion space tightness can adversely affect patellofemoral tracking and knee flexion.

## 1. Introduction

Tibiofemoral congruency is among the several features considered when designing total knee arthroplasty (TKA) inserts. Since the late 1990s, there has been a growing interest in closely mimicking the physiological medial-pivoting movement and the rollback of the lateral femoral condyle of the native knee, which, when combined, determine the internal axial rotation of the tibia relative to the femur during flexion [1,2,3,4]. Randomized trials and kinematic studies report clinical and biomechanical improvements using the medial-pivot design relative to low-congruent posterior cruciate ligament (PCL) retaining (CR), posterior-substituting (PS), and ultracongruent (UC) designs [5,6,7,8]. However, the consequences of adding different levels of congruency to the lateral insert are less clear, especially when there is excessive tension in the PCL that can overconstrain the flexion space which is detected by anterior lift-off of the trial insert from the tibial baseplate at 90° flexion [9,10].

Although there is a consensus that a TKA should restore native knee flexion and extension, of equal importance could be the restoration of native axial rotation as it changes the Q-angle during knee flexion, which determines retinacular ligament tension and patellofemoral tracking [2,11,12]. In the native knee, tibial rotation occurs about a medial pivot because of the medial compartment’s ball-in-socket-like congruency supplemented by the coronary ligament’s stabilizing effect of the medial meniscus and the lateral convex tibia with the mobile lateral meniscus that functions as a flat articular surface [3,11]. In addition, in the native knee and TKA, the PCL tension drives internal tibial rotation [9,11]. Therefore, a TKA with native knee congruency and an intact PCL could restore the external tibial orientation (i.e., screw-home) in extension and internal tibial orientation in 90° flexion, thereby promoting patellofemoral tracking.

In TKA, determining how the level of lateral insert congruency determines external and internal tibial orientation requires an insert with medial ball-in-socket congruency that mimics the morphology of the native knee and an intraoperative tool to measure it. A medial congruency less conforming than a ball-in-socket causes a loss of tibial rotation [3,13]. A medially inscribed radial dial converts the trial insert into a goniometer that can measure external tibial orientation at extension, which indicates the extent of the screw-home movement and the internal tibial orientation in 90° flexion relative to a sagittal line on the femoral trial component (Figure 1) [13].

There are no reports on the effect of changing the lateral insert’s level of congruency on tibial rotation and flexion space laxity when the medial insert has a ball-in-socket congruency and the PCL is retained. Accordingly, this study tested the hypothesis that a lateral flat insert maximizes external tibial orientation at extension and internal orientation at 90° flexion and lowers the incidence of anterior lift-off relative to low-congruent (LC) and ultracongruent (UC) lateral inserts (Figure 2).

## 2. Materials and Methods

Our institutional review board approved this retrospective study (IRB 00054838). Between April 2021 and June 2021, two surgeons trialed 3-D-printed, one-time-use goniometric inserts on a first-come, first-serve basis in 23 patients. The implant manufacturer, without cost, provided inserts with three different lateral sagittal congruencies based on the GMK Sphere (flat), GMK Primary CR (low-congruent), and GMK Primary Ultracongruent and a medial ball-in-socket congruency (Medacta, Castel San Pietro, Switzerland). The surgeons implanted the TKA with unrestricted caliper-verified kinematic alignment (KA) and PCL retention using manual instruments that accurately restore the patient’s prearthritic joint lines [14,15]. Each patient fulfilled the Centers for Medicare & Medicaid Services guidelines for medical necessity for TKA treatment including: (1) radiographic evidence of Kellgren–Lawrence Grade II to IV arthritic change or osteonecrosis; (2) any severity of clinical varus or valgus deformity; (3) and any severity of flexion contracture. Excluded were patients with prior intra-articular fracture, bone loss from avascular necrosis, and septic arthritis.

### 2.1. Overview of Unrestricted Caliper-Verified KA Technique and Accuracy Analysis of Component Placement

The following is an overview of the unrestricted caliper-verified KA technique performed through a midvastus approach with patella resurfacing using intraoperatively recorded verification checks and following a decision tree [16]. For the femoral component, the varus–valgus (V–V) and internal–external (I–E) orientations and the A–P and proximal–distal (P–D) positions were set coincident with the patient’s individual prearthritic distal and posterior joint lines by adjusting the thicknesses of the distal and posterior femoral resections measured by a caliper to within 0 ± 0.5 mm of those of the femoral component condyles, after compensating for cartilage wear and the kerf of the saw blade. The basis for setting the distal and posterior femoral resection guides is knowing that the varus and valgus Grade II to IV Kellgren–Lawrence osteoarthritic knees have negligible bone wear at 0° and 90° of flexion and that the mean full-thickness cartilage wear approximates 2 mm [17]. An accuracy analysis showed that these steps restored the distal lateral femoral joint line of 97% of patients within normal left to right symmetry and set the I–E orientation of the femoral component with a deviation of 0.3° (external) ±1.1° from the KA target of the F–E plane of the patient’s knee [18,19,20,21].

The surgeons followed six options in a decision tree to set the V–V orientation and posterior slope of the tibial component to restore the patient’s prearthritic tibial joint line and limb alignment and balance the knee by restoring the native tibial compartment forces [22,23,24]. The varus–valgus orientation of the proximal tibial resection was adjusted working in 1°–2° increments until there was negligible medial and lateral lift-off from the femoral component during a varus–valgus laxity assessment at extension with the spacer block and trial tibial insert. An accuracy analysis showed that these steps restored the proximal medial tibial joint line of 97% of patients within normal left-to-right symmetry [19,21,25]. The method for visually selecting the posterior slope was to set an angel wing, inserted through the tibial guide’s medial slot, parallel to the patient’s prearthritic slope (Figure 2). A three-dimensional accuracy analysis in osteoarthritic varus knees reported a mean difference of 0° ± 2.5° between the patient’s individual prearthritic and tibial component’s posterior slope [25]. A best-fit of the largest anatomically shaped trial tibial baseplate inside the cortical rim of the proximal tibial resection set the I–E orientation and A–P and medial–lateral (M–L) positions. An accuracy analysis showed a mean 2° (external) ± 5° deviation of the I–E orientation of the tibial component from the KA target of the F–E plane of the patient’s knee [18,19,26].

The surgeons inserted the trial femoral and tibial components and an insert that matched the thickness of the spacer block. With the knee in 90° flexion, the surgeons palpated the PCL to verify that it was intact. With the knee in extension, the surgeons verified that the knee hyperextended a few degrees, like the prearthritic knee. When the knee had a flexion contracture, either a thinner insert was placed, or the posterior capsule was released. The surgeons manually verified that V–V laxity was negligible in full extension and, with the knee in 15°–30° flexion, that the medial gap was negligible and the lateral gap was approximal 3 mm, like the native knee [27]. When necessary, the V–V plane of the tibial resection was fine tuned.

### 2.2. Method for Measuring the Orientation of the Tibia and Recording Anterior Lift-off of the Trial Insert

The surgeons opened a sterile package that matched the implanted insert thickness and contained three 3-D-printed trial goniometric inserts with each level of lateral insert congruency. An online randomization program (www.randomization.com accessed on 4 March 2021) determined the sequence for evaluating the different levels of lateral congruency. First, the surgeons reduced the patella, placed the patient’s heel on the back of their wrist, and lifted the leg to passively extend the knee in maximum extension without applying an I–E moment to the ankle. The insert goniometer measured the external tibial orientation relative to the femoral component (+ external/− internal) (Figure 1). Next, the surgeon placed the knee in 90° flexion, rested the foot on the operating table, and measured the internal tibial orientation. At 90° flexion, the surgeons checked for anterior lift-off of the insert from the trial baseplate. Repeating these steps evaluated the two other levels of lateral insert congruency. The surgeons recorded the measurements of tibial orientation with a thigh tourniquet inflated to 250 mm Hg.

### 2.3. Statistical Analysis

Data were analyzed using statistical software (JMP^®^ Pro 16.2.0, www.jmp.com, accessed on 28 March 2021), SAS, Cary, NC, USA). A single-factor repeated-measures analysis of variance with three levels (i.e., flat, low-congruent, and ultracongruent) determined whether there was a difference in mean tibial orientation in extension and at 90° flexion. For each analysis, a Tukey’s honest significant difference (HSD) post hoc test determined the differences between all pairs of insert congruency. Significance was *p* < 0.05.

To quantify the reproducibility of the goniometric measurement, two surgeons measured the external tibial orientation in extension and tibial internal orientation at 90° flexion in 7 randomly selected patients. Software computed the intraclass correlation coefficient (ICC) for each measurement using a 2-factor analysis of variance with random effects. The first factor was the observer with two levels (surgeons 1 and 2). The second factor was the 7 patients. An ICC value of >0.9 indicates excellent agreement, and 0.75–0.90 indicates good agreement [28]. ICC values of 0.82 for the measurement of the external tibial orientation in extension and 0.87 for internal tibial orientation at 90° flexion indicated good reproducibility.

The lack of historical differences in the tibial orientation from a repeated-measures study design precluded an a priori power analysis. Instead, the study included a post hoc computation (G*Power 3.1.9.6 for Mac OS 10.7 to 12). The inputs consisted of Type I error (alpha) of 0.05, power = 0.95, sample size of 23, number of patient groups = 1, number of external or internal orientation measurements = 3. The study had an effect size of 0.35. With an error variance of 3.6°, the corresponding sensitivity for detecting a difference in external and internal orientation was 0.7°, which is lower than the 1° resolution of the goniometer. Hence, the present study was adequately powered and could detect a 1° difference in tibial orientation between lateral insert congruencies.

## 3. Results

The study consisted of 23 patients, 39% females, a mean age at the time of surgery of 68 ± 10 years (48 to 89), and a mean BMI of 29 ± 5 kg/m^2^. Percentages and descriptive statistics of preoperative clinical characteristics, knee conditions, radiographic Kellgren–Lawrence arthritic rating, and function at the time of surgery are shown in Table 1.

In extension, the F insert’s mean external tibial orientation of 9° was greater than that of the LC (5°) (*p* < 0.0001) and the UC (2°) inserts (*p* < 0.0001) (Figure 3).

At 90° of flexion, the magnitude of the F insert’s mean −13° of internal tibial orientation was greater than the LC (−9°) (*p* < 0.0001) and the UC (−7°) inserts (*p* < 0.0001) (Figure 4).

At 90° flexion, the lateral flat insert’s 0% incidence of anterior lift-off was less than that of the LC (26%) and UC inserts (57%) (*p* < 0.0001).

## 4. Discussion

The most important findings of the present study were that increasing the lateral congruency from a flat articular surface caused a loss of the external tibial orientation in extension, a loss of internal tibial orientation at 90° flexion, and anterior lift-off of the insert, which indicated an over-tight PCL. Understanding the magnitude of the loss of tibial orientation and tightness in the flexion space caused by increases in the lateral anterior and posterior upslope might interest those responsible for implant design.

Increasing the anterior upslope of the lateral insert had a predictable “chock-block effect” on the external tibial orientation in extension. However, the previously unreported loss of 3° and 6° of screw-home rotation caused by the LC and UC lateral inserts, respectively, and its potential effect on patellofemoral tracking requires discussion. Restoring native external tibial rotation in extension could be functionally important as it assists in capturing the patella through the tensioning of the lateral and medial retinacula when it is not yet confined in the trochlear groove [29]. Hence, a lateral flat insert without an anterior upslope could promote a more physiologic screw-home and native patellofemoral kinematics in extension.

Similarly, increasing the posterior upslope of the lateral insert had a predictable “chock-block effect” on the internal tibial orientation in 90° flexion. However, the previously unreported loss of 5° and 7° of internal tibial orientation with the LC and UC lateral inserts and its potential effect on patellofemoral tracking requires discussion. In the native knee, the tibia internally rotates during flexion, which decreases the Q-angle and optimizes the retinacular ligament tension that guides patellofemoral tracking [2,11,12]. Hence, a lateral insert without a posterior upslope, which promotes internal tibial rotation, could potentially minimize the risks of lateral patellar displacement and anterior knee pain [29,30,31].

Anterior lift-off of the tibial insert indicates an over-tensioned PCL and a tight flexion space, which did not occur when trialing the lateral flat insert at 90° flexion [32]. In contrast, the posterior upslope of the LC and UC lateral inserts, which engages the femoral component in 90° flexion, caused anterior lift-off (Figure 2). Had the surgeon implanted an insert with lateral LC or UC, there would have been a need to release the healthy but over-tensioned PCL to correct the over-tight flexion space and reduce the risk of flexion loss. However, the insert with the lateral flat articular surface reduced the risk of the adverse consequences associated with PCL resection, which are increased laxity in the flexion space and tibiofemoral instability [33]. Accordingly, the lateral flat insert enabled PCL retention and lowered the risks of over-tensioning the PCL, stiffness, and limited knee flexion.

There are limitations to generalizing the results of the present study, which the reader should understand. First and foremost, although a flat lateral articular surface provided rotational and flexion-space balancing benefits, the present study’s design could not determine any clinical outcome and long-term implant survival benefits relative to LC and UC articular surfaces. Second, the reported tibial orientation measurements were the result of analyzing an insert with a medial ball-in-socket replicating the native knee geometry and the lateral articular congruencies tested. These orientation measurements do not apply to posterior stabilized, PCL retaining, and low-conforming insert geometries that are less constrained medially [2,3,7,34]. Third, the tibial orientation measurements and risk of anterior lift-off of a TKA implanted with mechanical alignment (MA) might be different and higher relative to caliper-verified KA, as MA changes the patient’s prearthritic joint lines. Fourth, the results are from a case series of two surgeons, and they require confirmation from others. Finally, the present study did not obtain pre- and postoperative measurements of the femoral and tibial mechanical axes because they are not used to perform or assess unrestricted caliper-verified KA. However, the wide radiographic range of preoperative knee deformity (i.e., 14° varus to −17° valgus) and the proportion of patients with medial (74%), lateral (22%), and patellofemoral (4%) osteoarthritis suggest a representative sampling of knee deformities, which enables the generalization of the effect of lateral insert congruencies on tibial orientation.

## 5. Conclusions

TKA surgeons and implant designers should be aware of the essential role that sagittal congruency determined by the anterior and posterior upslope of the lateral insert plays in restoring tibial rotation and retention of the PCL.

## Figures and Tables

**Figure 1 jpm-12-01274-f001:**
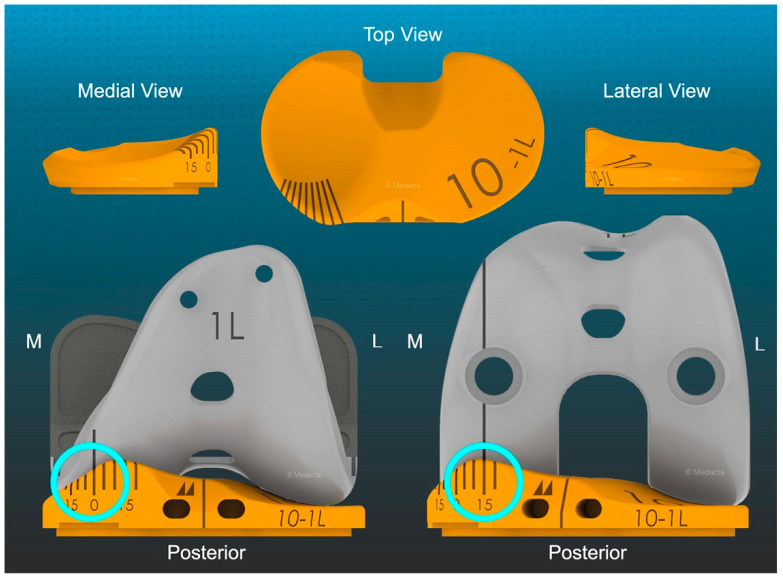
Schematics showing poses of the trial insert goniometer (top row) and the measurement of 0° external orientation or screw-home movement in extension and 15° internal tibial orientation at 90° flexion (cyan circles) relative to a laser etched sagittal line (black) on the femoral component (bottom row) (GMK Sphere, Medacta).

**Figure 2 jpm-12-01274-f002:**
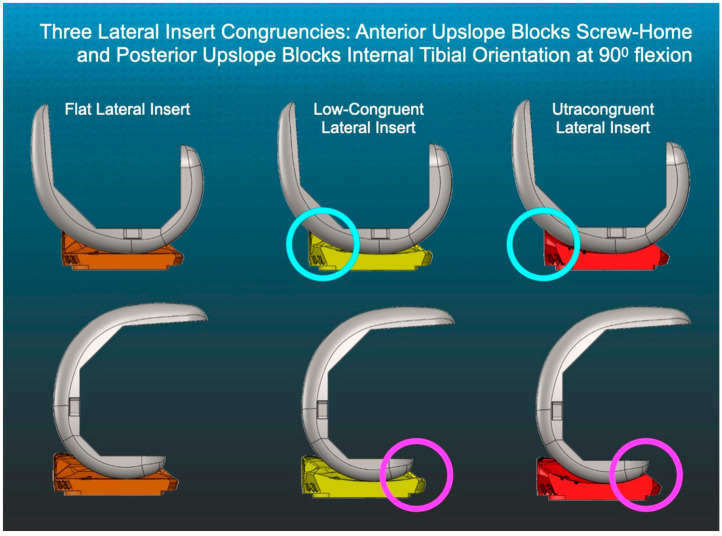
Schematics show the sagittal profiles of flat (F), low-congruent (LC), and ultracongruent (UC) lateral inserts relative to the femoral component with the knee in extension (top row) and at 90° flexion (bottom row).

**Figure 3 jpm-12-01274-f003:**
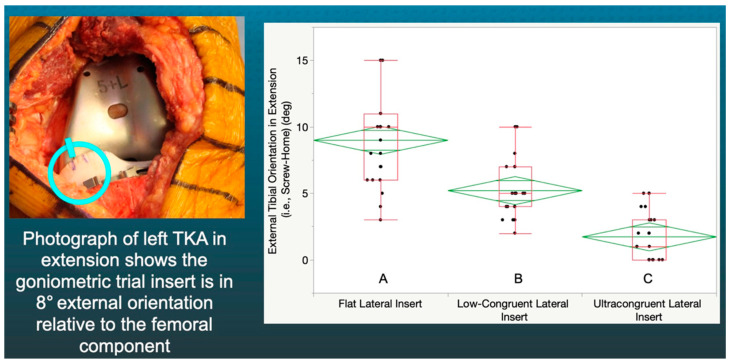
Box plot showing external tibial orientation in flexion for the three insert congruencies. Congruencies not connected by the same letters were significantly different.

**Figure 4 jpm-12-01274-f004:**
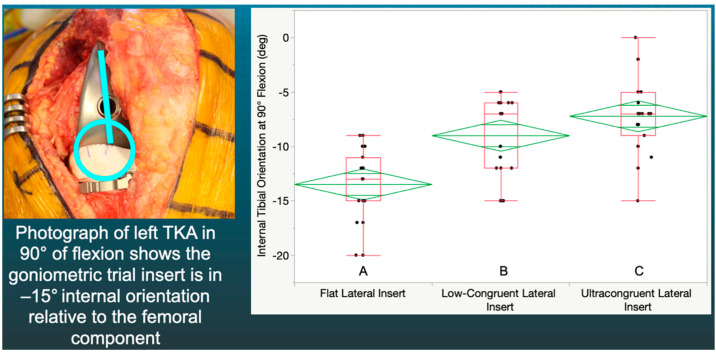
Box plot showing internal tibial orientation in 90° flexion for three insert congruencies. Congruencies not connected by the same letters were significantly different.

**Table 1 jpm-12-01274-t001:** Preoperative characteristics, knee conditions, and function scores for the 23 patients in the present study.

Preoperative Characteristics	Values ± SD (Range)
Age	68 ± 10 years (48 to 89)
Sex	9 females, 14 males
Body mass index	30 ± 5 kg/m^2^ (22 to 42)
Extension	7 ± 7° (0 to 24°)
Flexion	112 ± 8° (90 to 130°)
The compartment in which osteoarthritis predominated	74% medial, 22% lateral, 4% patellofemoral
Radiographic knee deformity (+varus, –valgus)	−1 ± 7° (14 to −17)
Kellgren–Lawrence classification	10% II, 50% III, 40% IV
Oxford Knee Score (48 is best, 0 is worst)	19 ± 6 points (7 to 28)
Knee Society Score (100 is best, 0 is worst)	35 ± 17 points (0 to 73)

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
