# Peer review of "A TKA Insert with A Lateral Flat Articular Surface Maximizes External and Internal Tibial Orientations without Anterior Lift-Off Relative to Low- and Ultracongruent Surfaces"

_jpm, 2022, doi:10.3390/jpm12081274_

Round 1
Reviewer 1 Report
I commend the authors on completing this interesting study. This study provides good information to the reader about implant designs.
Please explain rationale for the sample size. Please provide the readers with power analysis rationale if performed.
Please explain how these trial inserts were designed. These appear to be custom made inserts from Medacta. Was the making of them funded by Medacta? The lateral compartment LC and UC designs were based on what company’s insert? Why did the authors not use LC and UC designs for the entire insert, which would more closely replicate currently sold designs?
Reviewer 2 Report
Thank you for the opportunity to review the manuscript. The review process aims to assess the quality and ensure the article's reliability, completeness, and consistency. It is a way to improve your manuscript, and I hope you find my comments helpful.
The study is attractive. Congratulations.
Title and Abstract: adequate and correct.
Keywords: OK
Introduction:
The authors note that "the importance of restoring tibial rotation in TKA is unclear." Why do you need to do this study?
Methods:
Figure 1 shows whether screw-home can better understand sagittal line.
Figure 2 Would it be better if UC's cyano circles is a little bit to the right.
Data indicating whether the tourniquet is measured after release
Line 143: Is there an extra ","here?
Results:
Table 1 has an error, Type of Osteoarthritic Knee Deformity--> Do you change patellofemoral to Normal.
We suggest you can provide the patient's pre-operative femoral mechanical angle (FMA) and tibial mechanical angle (TMA) data.
Discussion:
Is the sample size of patients in this study too small?
Conclusion: OK
